# Enhanced stress resilience in potato by deletion of *Parakletos*

Muhammad Awais Zahid [1], Nam Phuong Kieu[1], Frida Meijer Carlsen [2], Marit Lenman[1], Naga Charan Konakalla [1], Huanjie Yang[1], Sunmoon Jyakhwa[1], Jozef Mravec[2,3], Ramesh Vetukuri[1,4], Bent Larsen Petersen [2], Svante Resjö[1] & Erik Andreasson [1] ✉

Continued climate change impose multiple stressors on crops, including pathogens, salt, and drought, severely impacting agricultural productivity. Innovative solutions are necessary to develop resilient crops. Here, using quantitative potato proteomics, we identify Parakletos, a thylakoid protein that contributes to disease susceptibility. We show that knockout or silencing of *Parakletos* enhances resistance to oomycete, fungi, bacteria, salt, and drought, whereas its overexpression reduces resistance. In response to biotic stimuli, *Parakletos*-overexpressing plants exhibit reduced amplitude of reactive oxygen species and $Ca^{2+}$ signalling, and silencing *Parakletos* does the opposite. Parakletos homologues have been identified in all major crops. Consecutive years of field trials demonstrate that *Parakletos* deletion enhances resistance to *Phytophthora infestans* and increases yield. These findings demark a susceptibility gene, which can be exploited to enhance crop resilience towards abiotic and biotic stresses in a low-input agriculture.

As a result of climate change, crops are likely to increasingly encounter biotic and abiotic stresses that negatively impact plant growth and yield that results in losses of billions of dollars[1]. To develop climate-change-resilient crops, it is important to understand plant responses to stress and identify factors that can provide tolerance to both biotic and abiotic stresses[2].

Plants have evolved sophisticated mechanisms to adapt to environmental changes[3]. These include complex signal transduction pathways involving reactive oxygen species (ROS), calcium ($Ca^{2+}$), and hormonal signalling, mediated by a network of receptors and other regulatory proteins[4]. A critical aspect of the plant immune response is pathogen-associated molecular pattern (PAMP)-triggered immunity (PTI), which relies on the perception of PAMPs via pattern-recognition receptors (PRRs)[5]. For example, Arabidopsis recognises flg22 (a 22-amino-acid epitope of bacterial flagellin) by PRR FLS2, which encodes a leucine-rich repeat receptor kinase (LRR-RK)[6]. This recognition activates parallel signalling pathways involving secondary messengers, such as ROS and $Ca^{2+}$, which trigger transcriptional changes[7].

Perception of biotic stimuli leads to biphasic ROS accumulation in two oxidative bursts. The first rapid ROS burst occurs within a few minutes after stress detection and is mainly generated by plasma-membrane-bound NADPH respiratory burst oxidase homologs (RBOHs)[8]. The second ROS burst, mainly contributed by the chloroplasts, is initiated within hours of stress perception[9,10]. $Ca^{2+}$ bursts help to propagate intracellular signals and are essential for PAMP-induced ROS production[7,11]. Chloroplast-mediated signalling pathways play key roles in generating ROS and $Ca^{2+}$ bursts; biosynthesis of phytohormones, such as salicylic acid (SA); and retrograde signaling[9,11]. The thylakoid-membrane-associated calcium-sensing receptor (CAS) plays a crucial role in these processes, including PTI-induced transcriptional induction, SA biosynthesis, and both bacterial as well as fungal resistance[12–14]. For example, pathogen

[1]Department of Plant Protection Biology, Swedish University of Agricultural Sciences, 234 22 Lomma, Sweden. [2]Department of Plant and Environmental Sciences, University of Copenhagen, Frederiksberg C, Denmark. [3]Present address: Institute of Plant Genetics and Biotechnology, Plant Science and Biodiversity Center,-Slovak Academy of Sciences, Akademická 2, 950 07 Nitra, Slovakia. [4]Present address: Department of Plant Breeding, Swedish University of Agricultural Sciences, 234 22 Lomma, Sweden. ✉e-mail: erik.andreasson@slu.se

effectors can modulate CAS activity in chloroplasts to suppress plant immune responses[15]. CAS also plays an important role in plant responses to abiotic stimuli[14]. Expression of rice CAS increases drought stress and preventing drought-induced reduction of photosynthetic efficiency in Arabidopsis [16].

It is becoming evident that immune signalling and abiotic stress responses overlap[17,18]. There is a need to elucidate the factors controlling biotic and abiotic stress signalling in plants and to identify factors that may confer broad-spectrum resistance to develop sustainable control strategies for agriculture. One such strategy, whose practical implementation in agriculture has been aided by advances in CRISPR technology[19], involves disrupting the expression of susceptibility (S) genes, i.e., plant genes that facilitate pathogen infection[20].

Here, to assist in the improvement of crop resilience, we examine the immune response of the agriculturally important crop potato by performing quantitative proteomic analysis. We identify a protein (dubbed Parakletos, an ancient Greek word meaning helper, which is chosen because the protein itself has no obvious functional domains), and demonstrate its importance in plant stress responses. Its overexpression in *Nicotiana benthamiana* reduces the intensity of flg22-induced ROS bursts and $Ca^{2+}$ bursts, whereas its silencing has the opposite effect, inducing over-expression of several defence-related genes after stress. Moreover, *parakletos*-knockout (KO) potato lines and *parakletos*-silenced *N. benthamiana* plants show increased resistance to several pathogens and improved tolerance to salt and drought stresses. Overall, our results indicate that Parakletos is a negative regulator of plant stress responses. In addition, *Parakletos* homologs are identified in all main crop plants, including wheat, rice and maize, suggesting that knocking it out could be a valuable tool for developing more resistant crops.

## Results

### Changes in abundance of potato proteins

In order to find breeding targets towards stress mitigation, we analysed changes in protein abundance in potato leaves in response to treatment with disarmed *Agrobacterium tumefaciens* (as compared to a control treatment with infiltration medium only), using a precursor intensity-based proteomics analysis (Supplementary Fig. 1a). After subcellular fractionation to reduce RuBisCo content, we identified and quantified 2178 proteins. Of these, 114 proteins increased significantly ($p < 0.05$) in abundance, and 51 decreased in abundance (Supplementary Data 1). Table 1 shows the 10 proteins exhibiting the highest changes in abundance. The proteins that increased in abundance included a plastocyanin and photosystem II reaction center protein L (PSII-L), and proteins decreased in abundance included MAPKs, histone H2B, and a hexose transporter.

### Functional test of candidate proteins identifies Parakletos

To assess the functional significance of the above findings, we analysed our dataset to identify potential immune response proteins. Five candidates were selected based on the criteria that each exhibited pronounced changes in abundance following treatment with disarmed *A. tumefaciens* and small gene family size. Homologs of the candidates in *N. benthamiana* were then identified. To assess their importance in plant immunity, the candidates were either transiently expressed or silenced in *N. benthamiana* plants that were subsequently infected with *Phytophthora infestans* (Table 2). Of the candidates selected for functional validation was a small protein M1CUF4, which was named as Parakletos. Overexpression of *Parakletos* increased *P. infestans* infection severity in terms of lesion diameter, sporangia counts, and increased *P. infestans* biomass, while its silencing showed the reversed phenotype (Fig. 1a, b and Supplementary Fig. 1b–d). Parakletos showed no significant homology to any known domains or proteins, and has no functional

## Table 1 | List of top ten proteins with the most significant increases and decreases in abundance

| Protein ID | Protein names | Fold change (log²) |
|---|---|---|
| M1A079 | Uncharacterized protein | 5.6 |
| M1ASJ0 | Homeobox-leucine zipper protein | 4.7 |
| M1CUF4 | Uncharacterized protein | 4.1 |
| M1BD66 | Stem 28 kDa glycoprotein | 3.3 |
| A0A0C4B0N5 | Uncharacterized protein | 3.2 |
| Q27S35 | Photosystem II reaction center protein L | 3 |
| M1B1N3 | 3-isopropylmalate dehydratase | 2.9 |
| M1ACB9 | 4F5 family protein | 2.8 |
| M1D3E9 | 4F5 domain containing protein | 2.8 |
| P00296 | Plastocyanin | 2.8 |
| M0ZWH0 | 5′–3′ exoribonuclease | −6.2 |
| M1BDT7 | NAD_Gly3P_dh_N | −4.6 |
| M0ZR27 | Mitogen-activated protein kinase | −2.5 |
| M1B1N5 | Chlorophyll a-b binding protein | −2.4 |
| Q2XPW1 | Histone H2B | −2.3 |
| M1CN16 | Transcription factor RF2b | −2.2 |
| M1D4R5 | Hexose transporter | −2.1 |
| M1C8J8 | 3-oxoacyl-CoA synthase | −2 |
| M1BS09 | Poly(RC)-binding protein | −2 |
| M0ZNA1 | Pre-mRNA-splicing factor cwc23 | −1.9 |

## Table 2 | Proteins screened for involvement in *Phytophthora infestans* resistance using transient expression or silencing in *Nicotiana benthamiana*

| Protein name | VIGS | Over-expression (OE) |
|---|---|---|
| Uncharacterized protein (M1CUF4-Parakletos) | Less infection | More infection |
| Transcription factor RF2b | NA | No difference |
| Hexose transporter | NA | No difference |
| Mitogen-activated protein kinase | No difference | NA |
| 5′–3′ exoribonuclease | No difference | NA |

data described in any system. However, localisation prediction algorithms[21] suggested that Parakletos contains both a chloroplast transfer peptide and a thylakoid luminal transfer peptide (Fig. 1c). To confirm the predicted subcellular localisation of Parakletos, we constructed a Parakletos C-terminal fusion with mCherry and transiently expressed it in *N. benthamiana*. As expected, immunoblotting after thylakoid-fractionation indicating localisation in the thylakoid (Fig. 1d) and confocal imaging showed that Parakletos was only localised in chloroplasts (Fig. 1e). Furthermore, confocal microscopy and immunoblotting experiments indicated a rapid and transient reduction in *Parakletos* protein levels following flg22 treatment in *N. benthamiana* and potato (Fig. 1e, f and Supplementary Fig. 1e, f). A decline in *Parakletos* protein was also observed post *P. infestans* inoculation, followed by an increase (Supplementary Fig. 1g). Additionally, by immunoblotting, an elevation in *Parakletos* protein levels was detected 18 hours after *Agrobacterium* infiltration (Supplementary Fig. 1h), corroborating the proteomic data. The three mass spectrometry peptides of Parakletos from the proteomics data and the conservation of the mature protein in the four potato alleles and *N. benthamiana* in Parakletos are shown in Fig. 1c. An alignment of sequences homologous to Parakletos indicates that all major crop plants examined contain this gene (Fig. 1g and Supplementary Fig. 1i).

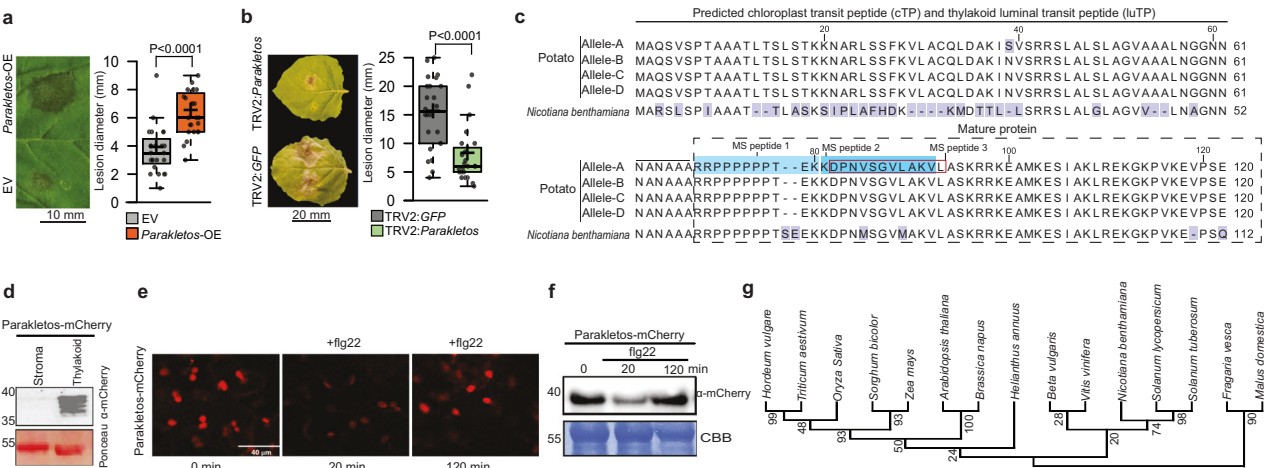

**Fig. 1 | Disease assay, sequence analysis and localization studies of Parakletos.** **a** *Phytophthora infestans* infection assay on *Nicotiana benthamiana* leaves infiltrated with disarmed *Agrobacterium* carrying Parakletos overexpression (*Parakletos*-OE) vector or empty vector (EV). **b** *N. benthamiana* leaves post-silencing using virus-induced gene silencing (VIGS) with TRV2:*GFP* or TRV2:*Parakletos* constructs. Exact *p*-values determined by two-tailed Student's *t*-test in **a, b** compared to control plants; box plots shown (*n* = 24, n=biological replicates). The centreline in box plots indicate medians, the + sign indicates the mean, the box borders delimit the lower and upper quartiles, and the whiskers show the highest and lowest data points. **c**, Sequence of Parakletos protein, including all four Désirée alleles, aligned with *N. benthamiana* homolog. **d** Immunoblot analysis showing localisation of Parakletos-mCherry fusion protein. **e** Confocal microscopy images of Parakletos-mCherry in *N. benthamiana* leaves at 0, 20, and 120 minutes post-treatment with flg22 (1 μM). **f** Immunoblot analysis of total protein (Parakletos-mCherry) in *N. benthamiana*, with and without flg22 treatment (1 μM at 0 h, 20 min, and 120 min). **g** Phylogenetic tree of putative plant Parakletos proteins, constructed using MEGA 11 software based on full-length sequences. All experiments were replicated thrice with similar results. Source data are provided as a Source Data file.

## Parakletos negatively regulates ROS, calcium bursts and defence-related gene transcription

The silencing of *Parakletos* and the transient decline in Parakletos protein abundance following flg22 and *P. infestans* treatment triggered us to test whether the temporary suppression of Parakletos may coincide with the accumulation of secondary messengers in the signalling pathway. To investigate this phenomenon, we monitored changes in the concentrations of two primary secondary messengers, ROS and $Ca^{2+}$, in *N. benthamiana* plants where *Parakletos* was either overexpressed or silenced, following flg22 treatment. *Parakletos* overexpression significantly weakened the amplitude of the first flg22-induced ROS burst (Fig. 2a), while its silencing enhanced it (Fig. 2b). Flg22 induction and ROS assays showed that expression of *Parakletos*-mCherry resulted in weaker ROS bursts than in control (mCherry-Empty vector [EV]) plants, confirming that the fusion protein was active (Supplementary Fig. 2a). A second ROS burst initiated after about 120 minutes, was also stronger in *Parakletos*-silenced plants than in controls (Fig. 2c). Moreover, expression of the *RBOHB* gene, which encodes a ROS-generating enzyme, increased in TRV2:*Parakletos* plants (Fig. 2d). Similarly, the flg22-induced $Ca^{2+}$ bursts were less intense in *Parakletos*-overexpressing plants and more intense in *Parakletos*-silenced plants (Fig. 2e, f). These findings suggest that Parakletos is a suppressor of immunity-related ROS and $Ca^{2+}$ signalling.

Since responses to *P. infestans* and flg22 treatment were linked to Parakletos, ROS production, and $Ca^{2+}$ levels, we examined changes in the expression of defence-related genes, including *isochorismate synthase 1* (*ICS1*), which is involved in SA biosynthesis[22], and *pathogenesis-related gene 1* (*PR1*), a common marker for the SA signalling pathway[23]. Treatment with flg22 increased *ICS1* and *PR1* expression in TRV2:*Parakletos* plants but had no significant effect on their expression in plants overexpressing *Parakletos* (Fig. 2g,h). Silencing or over-expressing *Parakletos* did not influence *ICS1 or PR1* expression without flg22 (Fig. 2g, h). Additionally, transcript levels of *PTI5*, a marker for PAMP-induced responses, increased following flg22 treatment in TRV2:*Parakletos* plants relative to controls (Fig. 2i). These results show that flg22 treatment triggered over-induction of several defence-related genes in TRV2:*Parakletos* plants, which is consistent with their reduced *P. infestans* infection severity and enhanced ROS and $Ca^{2+}$ bursts.

After observing the significant effects of Parakletos manipulation in *N. benthamiana* i.e. its role in enhancing pathogen resistance and its involvement in stress signalling, we moved back to potato. We aimed to further investigate its role in crop plants and to explore the potential for field trials. To achieve this, we generated *parakletos* knockout (KO) lines in potato using CRISPR/Cas9 technology. We confirmed full tetra-allelic knockout of *parakletos* in these lines by Sanger sequencing analyses (Fig. 2j and Supplementary Fig. 3). These KO lines exhibited enhanced ROS bursts like those seen in *parakletos*-silenced *N. benthamiana* plants (Fig. 2k and Supplementary Fig. 2b), and NBT staining experiments revealed that their superoxide ($O_2^{\cdot-}$, a major ROS) levels increased only after *P. infestans* inoculation (Fig. 2l and Supplementary Fig. 2c). Moreover, microscopic examination of sections from NBT-stained leaves showed increased staining in the chloroplasts of *Parakletos*-KO lines, indicative of elevated chloroplastic ROS (cROS) (Supplementary Fig. 2d). Furthermore, overexpression of Parakletos-mCherry in potato resulted in less ROS (Supplementary Fig. 2e). These data indicate that the function of Parakletos is conserved between *N. benthamiana* and Potato.

We next tested the hypothesis that Parakletos acts in the same pathway as CAS, a thylakoid master regulator of plant immune responses, by silencing their expression individually and jointly (Supplementary Fig. 2f). The resulting plants were then subjected to ROS and $Ca^{2+}$ assays following flg22 exposure and *P. infestans* infection (Fig. 2m–o). In accordance with previous described results (Fig. 2b, c), *Parakletos* silencing enhanced the flg22-induced first and second ROS bursts. Interestingly, in plants in which both *Parakletos* and *CAS* were silenced, the ROS bursts were not significantly stronger than those in TRV2:*GFP* control or TRV2:*CAS* plants (Fig. 2m) but were clearly weaker than in plants in which only *Parakletos* was silenced. Similar results were found for $Ca^{2+}$ assay (Fig. 2n). Also, *P. infestans* infection severity was reduced in TRV2:*Parakletos* plants compared with TRV2:*GFP* controls, and infection severity in TRV2:*Parakletos/CAS* and TRV2:*CAS* plants did not differ significantly from controls (Fig. 2o). Co-immunoprecipitation experiments showed that Parakletos interacts with CAS, but not with the negative control CPK16$^{G2A}$, a mutant which localizes in chloroplasts[12] (Supplementary Fig. 2g). These findings suggest that Parakletos and CAS are in the same protein complex.

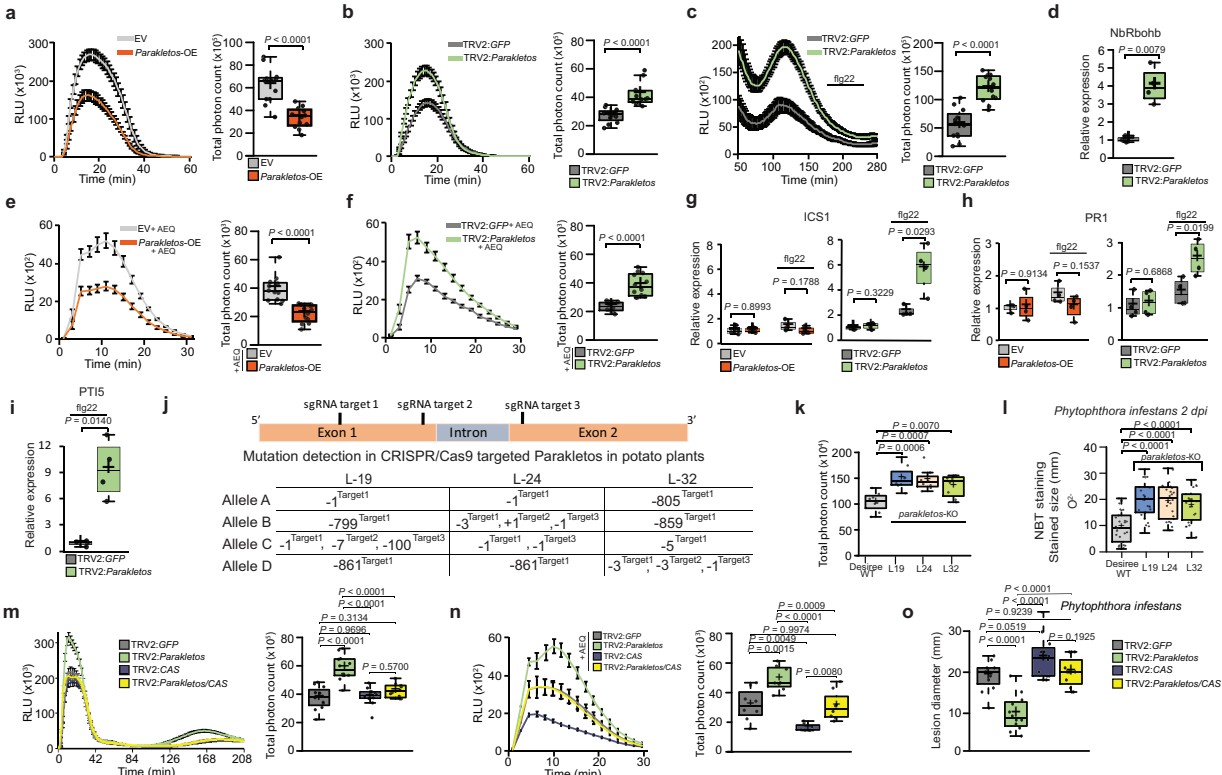

**Fig. 2 | Parakletos negatively regulates plant immune signaling. a, b** ROS production in response to flg22 (1 μM) in *Nicotiana benthamiana* plants transiently expressing Parakletos, or in silenced plants, measured as total relative light units (RLU). **c** Elevated second ROS peak in response to 0.1 μM flg22 in *Parakletos*-silenced plants. **d** Expression of *NbRBOHB* (normalized to EF-1α) in *Parakletos*-silenced plants after infiltration (12 hpi) with flg22 (1 μM) relative to controls. **e, f** Aequorin (AEQ) was transiently expressed along with Parakletos for over-expression studies or transiently expressed alone in 3-4 week-old silenced plants to monitor cytosolic $Ca^{2+}$ bursts in response to flg22 (1 μM). **g–i** Relative expression levels of (**g**) *ICS1* (12hpi), (**h**) *PR1*(12 hpi) and (**i**) *PTI5* (45 minutes post infiltration) (normalized to EF-1α) were measured in *N. benthamiana* leaves in which *Parakletos* was transiently expressed or silenced, with or without flg22 treatment. **j**, Tetraallelic detection of mutations sizes in CRISPR/Cas9 generated *parakletos*-KO (knock-out) potato lines (L19, L24, and L32). **k**, ROS production during the response to flg22 (8 μM) in *parakletos*-KO potato plants and Désirée-WT (background control).

**l** *parakletos*-KO potato plants and Désirée-WT were subjected to nitroblue tetrazolium (NBT) staining 2 days post-infection with *Phytophthora infestans*. **m–o** Effects of dual Parakletos and CAS silencing in *N. benthamiana*. **m**, ROS levels determined in 3–4-week-old silenced plants induced with 1 μM flg22. **n**, Cytosolic $Ca^{2+}$ bursts in response to flg22 (1 μM). **o**, Lesion diameter (mm) in the *Phytophthora infestans* infection assays. Data were analyzed by two-tailed Student's *t*-tests (**a–i, k,l**) or one-way ANOVA followed by Tukey's test (**m–o**). Exact *p* (**a–i, k,j**) or adjusted *p* (**m-o**) values were shown in figures. Box plots display individual data points for **a–i, k–o** (**a–c, e, f**, *n* = 12; **d**, *n* = 4; **d, g–l**, *n* = 4; **k**, *n* = 8; **l**, *n* = 24; **m**, *n* = 12; **n**, *n* = 8; **o**, *n* = 12, *n*=biological replicates). The centerline in box plots indicate medians, the + sign indicates the mean, the box borders delimit the lower and upper quartiles, and the whiskers show the highest and lowest data points. All experiments were conducted at least two to three times with similar results. Source data are provided as a Source Data file.

## Absence of Parakletos enhances broad-spectrum disease resistance and increases salt and drought stress tolerance

As Parakletos caused changes in conserved stress signalling pathways via ROS and $Ca^{2+}$, we further investigated the effects of various stress conditions on *parakletos*-KO potato mutants. These mutants were less severely infected by both *P. infestans* (hemibiotroph) and *Alternaria solani* (necrotroph) than WT controls (Fig. 3a, b). We also performed disease assays using the bacterial pathogens *Dickeya dantatii* (necrotroph) and *Pseudomonas syringae pv. tomato DC3000 HopQ1* (hemibiotroph) in *Parakletos*-silenced *N. benthamiana* plants. Again, the degree of infection in *Parakletos*-silenced plants was lower than in controls (Fig. 3c, d). Finally, salt and drought stress experiments showed that *parakletos*-KO potato mutants are more tolerant to these stress conditions than controls (Fig. 3e, f and Supplementary Fig. 4a–c). We also detected no significant differences between *parakletos* mutants and Désirée-WT controls in terms of plant height, biomass, or other visible traits (Supplementary Fig. 4d). These results clearly show that *Parakletos* loss of function increases plant resilience to both biotic and abiotic stress without affecting plant growth under controlled conditions.

## CRISPR/Cas9 deletion of Parakletos enhances late-blight resistance and yield in the field

Potato late blight, caused by *P. infestans*, is a major disease that can severely reduce potato yield, and requires intense fungicide treatment to eradicate it. Therefore, after testing *Parakletos* mutants under controlled conditions, we assessed their growth and resistance to *P. infestans* in open GM-classified field trials (with no fences) conducted in southern Sweden (Fig. 4a). *Parakletos* mutant lines (designated *parakletos*-19 and −24) and Désirée-WT controls were cultivated in a random block design with four replicates. We found that the severity of late blight in *Parakletos* mutant lines was significantly lower than those of the Désirée-WT controls, as indicated by the area under the disease progress curve (AUDPC) (Fig. 4b, c). In both years of testing, the KO lines showed at least 20% increases in yield compared with Désirée-WT (Fig. 4c) in plots without fungicide treatments against *P. infestans*. Furthermore, we observed no significant differences in growth parameters or visible traits between *Parakletos* mutant lines and Désirée-WT controls, or in other parameters such as quantum yield of PSII (φPSII) and stomatal conductance (mmol m$^{-2}$ s$^{-1}$) (Fig. 4d and Supplementary Fig. 4e–g). In parallel trials at the same location,

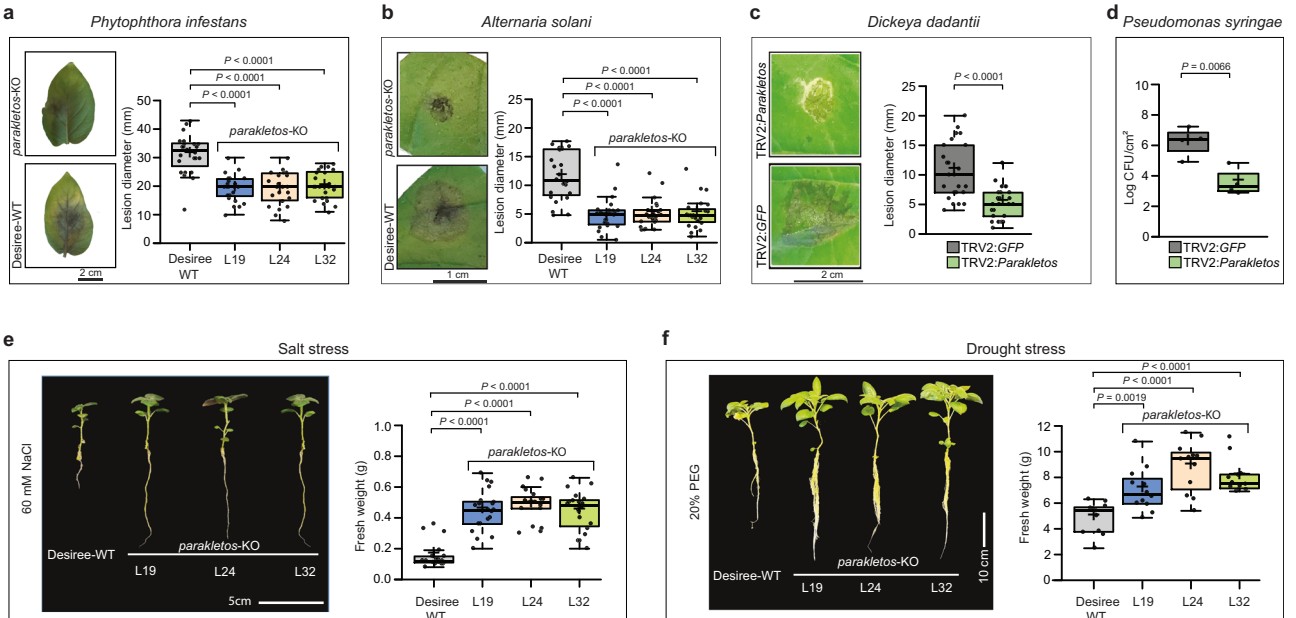

**Fig. 3 | *Parakletos*-KO and *parakletos*-silenced plants exhibit broad-spectrum disease resistance and enhanced salt and drought stress tolerance. a–d**, Disease symptoms and infection assays with (**a**) the oomycete pathogen *Phytophthora infestans*, (b) the fungus *Alternaria solani* on *Parakletos*-KO potato plants, and (**c,d**) the bacterial pathogens *Dickeya dadantii* and *Pseudomonas syringae* dHopQ in *Nicotiana benthamiana* parakletos-silenced plants. **e**, Effect of salt stress (60 mM NaCl) on *parakletos*-KO plants. **f** Effect of drought stress (20% PEG) on *parakletos*-

KO plants. Individual data points are plotted as box plots in **a–c** ($n = 24$), **d** ($n = 4$), **e** ($n = 20$) and **f** ($n = 12$), $n$=biological replicates. The centerline in box plots indicate medians, the + sign indicates the mean, the box borders delimit the lower and upper quartiles, and the whiskers show the highest and lowest data points. Exact $p$-values were determined using two-tailed Student's $t$-tests on treated plants compared with control plants. Experiments were repeated at least three times with similar results. Source data are provided as a Source Data file.

but with standard efficient fungicide treatments (no visible diseases), no significant yield differences were detected between Désirée-WT and the KO lines (Fig. 4e). These findings indicate that knocking out *Parakletos* increases resistance to *P. infestans* resulting in yield increase with no apparent effect on growth and that Parakletos can be a tool for future sustainable agriculture.

## Discussion

A quantitative proteomic analysis of proteins from potato leaves identified over 100 proteins with abundances that were significantly affected by challenge with disarmed *A. tumefaciens* (Supplementary Data 1), indicating their potential involvement in general plant immune and stress responses. Many of the differentially regulated proteins in our proteomics dataset are related to photosynthesis, which is consistent with the idea that chloroplasts play important roles in coordinating biotic and abiotic stress responses[24]. In addition to photosynthesis, chloroplasts contribute to plant stress resilience by synthesizing hormones, such as SA, and harbouring secondary signalling messengers, including ROS and $Ca^{2+}$ [25].

Screening of candidate proteins in our proteomics dataset revealed a thylakoid protein that we named Parakletos. We showed that Parakletos is a negative regulator of resistance to diverse pathogens, including *P. infestans* (Figs. 1a, b and 4a–d). Over-expressing *Parakletos* in *N. benthamiana* increased the severity of infection, while silencing it enhanced resistance; these traits were associated with changes in the amplitude of ROS and $Ca^{2+}$ bursts (Fig. 2a–f). ROS production has been linked to resistance to multiple pathogens, and perception of plant pathogens triggers biphasic ROS bursts[8]. *Parakletos* silencing enhances the first flg22-induced ROS burst and increases resistance to *P. infestans* infection. Recently, the gene *MKK1* has been shown to negatively regulate PTI by suppressing flg22-induced ROS bursts, thus promoting *P. infestans* infection[26]. *Parakletos* silencing also increased the transcription of several defence-related genes after flg22 treatment. One such gene, *RBOHB*, was more strongly

expressed in *Parakletos*-silenced plants than in controls (Fig. 2d), suggesting a mechanism by which Parakletos could affect plasma-membrane-related ROS production. However, elevated *RBOHB* transcript levels are unlikely to affect flg22-induced ROS bursts directly because transcript regulation occurs on a longer timescale than ROS bursts. Besides affecting flg22-induced ROS bursts associated to the plasma membrane, *Parakletos*-KO plants also exhibited increased cROS following *P. infestans* inoculation (Supplementary Fig. 2d). The chloroplast is a main site of ROS production in plant cells and plays an important role in redox homeostasis and retrograde signalling[27]. We also found that expression of the SA-biosynthesis-related gene *ICS1*, and the SA signalling pathway marker gene *PR1*, is increased in *Parakletos*-silenced plants (Fig. 2g, h). This is consistent with previous reports that SA levels are related to broad-spectrum disease resistance[28], as well as potato resistance to necrotrophic pathogens such as *A. solani* and *Dickeya solani*[29,30]. Flg22 treatment also increased the expression of a PTI marker gene *PTI5* in *Parakletos*-silenced plants (Fig. 2i). PTI5 participates in regulation of ROS- and SA-regulated gene expression[31]. Furthermore, elevated SA levels promote ROS accumulation in accordance with our detection of elevated levels of transcripts of both ROS- and SA-related genes in *Parakletos*-silenced plants following biotic stress[32,33].

The effects of *Parakletos* silencing on $Ca^{2+}$, ROS, and *P. infestans* resistance in *N. benthamiana* were reversed if CAS was also silenced (Fig. 2m–o). Additionally, we present an indication that CAS and Parakletos may function together within a complex (Supplementary Fig 2g). Therefore an attractive hypothesis is that Parakletos inhibits CAS function during biotic and abiotic stress, and that the transient reduction in Parakletos levels we observed relieves this repression.

We show that CRISPR/Cas9-mediated KO of *parakletos* in potato or silencing of *Parakletos* in *N. benthamiana* provides broad-spectrum resistance to *P. infestans, A. solani, D. dantanii,* and *P. syringae* and enhances plant tolerance towards salt and drought stress under

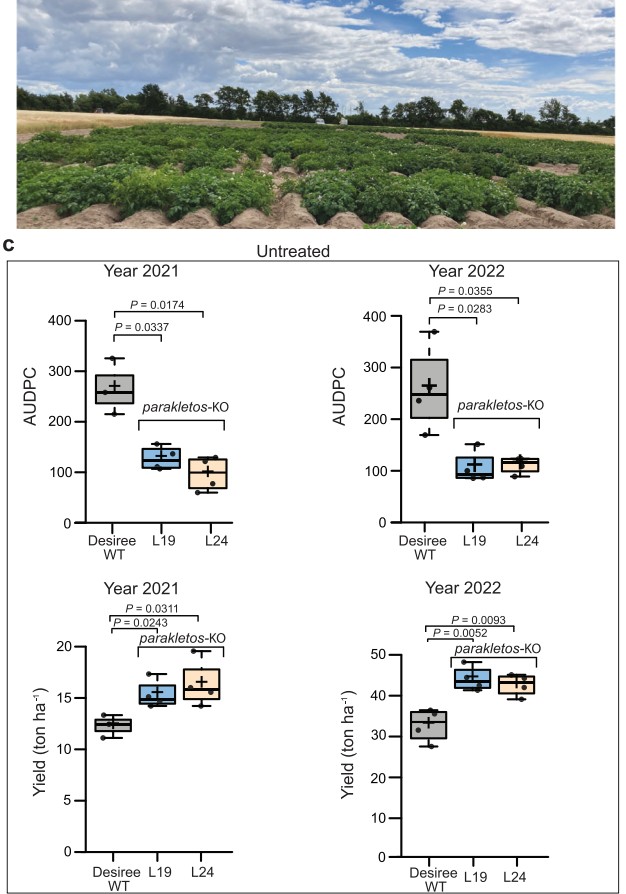

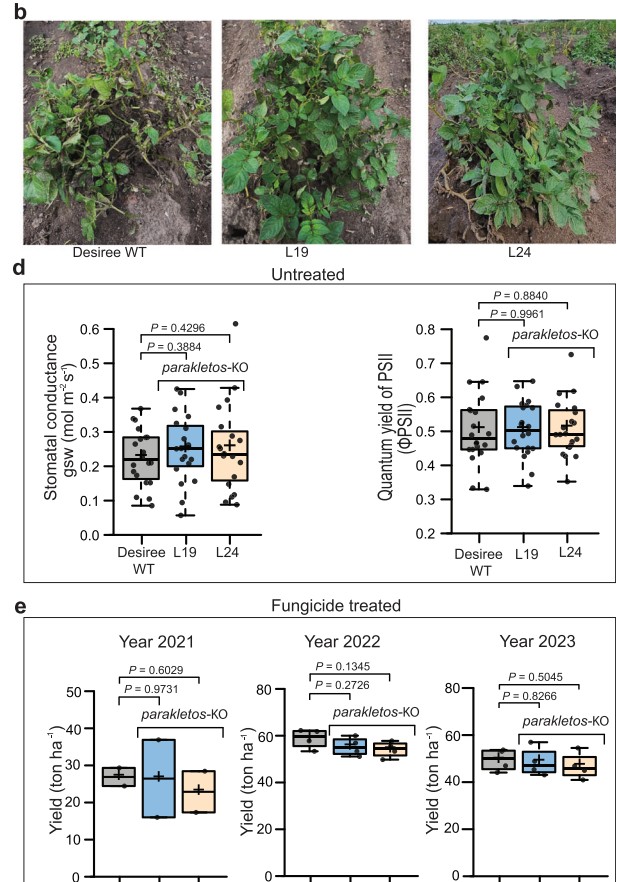

**Fig. 4 | *Parakletos*-KO lines showed enhanced resistance to late blight and increased yield in the field. a** Photograph of potato plants in the field trial at Borgeby, Sweden. **b**, Representative image of plants naturally infected with *Phytophthora infestans* in the field. **c** Untreated plot for late blight, disease incidence based on area under the disease progress curve (AUDPC) for *parakletos*-KO lines (L19 and L24) compared with Désirée background control for year 2021 and year 2022 and in lower panel tuber yield in tonnes per hectare (ton ha⁻¹) in 2021 and 2022. **d** Physiological parameters of indicated lines in untreated plot for the year 2022: quantum yield of PSII (φPSII) and Stomatal conductance (mmol m⁻² s⁻¹). **e** For

fungicide-treated plots, tuber yield in tonnes per hectare (ton ha⁻¹) obtained at in 2021, 2022, and 2023. Individual data points are plotted as box plots in **c**–**e**, with (**c**) year 2021 (Désirée =3, L19 = 4, L24 = 4), year 2022 (*n* = 4), (**d**) *n* = 20, and (**e**) year 2021 *n* = 2, year 2022 *n* = 4, year 2023 *n* = 4. The centerline in box plots indicate medians, the + sign indicates the mean, the box borders delimit the lower and upper quartiles, and the whiskers show the highest and lowest data points. Data were analysed by two-tailed Student's *t*-test compared with respective control plants, exact *p*-values were shown in figures. Source data are provided as a Source Data file.

controlled conditions (Fig. 3). This is consistent with the effects of other genes that affect the ROS, Ca²⁺, and SA pathways[28,34]. However, mitigation of salt and drought stress has not to our knowledge been directly shown with other S genes. Deleting *parakletos* did not cause a significant growth penalty, and no changes in expression of defence genes were detected in unchallenged plants; thus, we hypothesized that Parakletos only functions under stress conditions, making it a good candidate for use in agricultural contexts. This might make *Parakletos* a more attractive susceptibility gene than for example *DMR6*, which exhibits increased defence gene expression in unchallenged KO plants[28]. Two years of field trials showed that CRISPR/Cas9-mediated *parakletos*-KO potato plants were significantly more resistant to late-blight disease than controls, increased yield by at least 20 %, and exhibited no detectable growth defects (Fig. 4 and Supplementary Fig. 4e–g). We emphasize that while *parakletos* deletion could contribute to late blight protection, it should be considered part of a broader integrated pest management strategy, rather than a standalone solution. This approach not only can extend the effectiveness of existing resistance genes but also provides a valuable defense against other stresses where options for fungicides or resistant genes are limited, including abiotic stress conditions.

A speculation could be that in nature, with a higher degree of competition than in agricultural settings, Parakletos is important in fine-tuning stress responses and thereby reducing the cost of resistance. In contrast, in an agricultural setting, with generally more light and nutrients for each plant, increased transient defence activation could be beneficial, for example, by the mutation of the negative regulator Parakletos. This is in line with the removal of deleterious mutations, which is an important factor in plant domestication[35]. To determine the full agricultural value of *Parakletos*, field trials should be repeated in several potato genotypes and locations. Also, CRISPR KO experiments targeting *Parakletos* homologs should be conducted in other crops (Fig. 1g).

In conclusion, our crop-to-model-to-crop study based on proteomics data identified a defence repressor protein named Parakletos. Parakletos may be classified as a susceptibility gene whose silencing or KO confers broad-spectrum resistance to multiple pathogens and enhances salt and drought stress tolerance without affecting plant growth. Moreover, knocking out this gene using CRISPR/Cas9 conferred resistance to *P. infestans* in field-grown plants and increased the yield at least 20%. These results illustrate the potential of crop proteomic analysis of general plant immunity in the search for factors that

can increase stress resilience and provide valuable tools for low-input agriculture.

## Methods

### Plant material and growth conditions

*Nicotiana benthamiana* and *Solanum tuberosum Désirée* plants were cultivated at 20 °C with 14/10 hour light/dark cycles in a controlled environment chamber. The light intensity was kept at 160 µmol m$^{-2}$ s$^{-1}$, and humidity at 65%. Plantlets of *N. benthamiana* were transplanted into separate pots two weeks after germination and grown for 3-4 weeks. Potato plantlets were transplanted into separate pots and grown for 3-5 weeks.

### Plasmid constructs and Parakletos gene sequencing

For over-expression studies candidate genes were amplified by PCR using primers with GATEWAY-compatible attB1 and attB2 tails and recombined, using BP clonase, into pDONR_201 (Life Technologies). Sequenced and selected pENTRY-clones were recombined with the destination vector pK2GW7.0, for gene expression driven by the 35 S CaMV promoter, using LR clonase (Life Technologies). For virus-induced gene silencing (VIGS) studies, the SGN VIGS Tool (https://vigs.solgenomics.net/) was used to select the best target region of each gene. An approximately 300 bp long region of each gene was amplified with PCR using primers containing BsaI overhangs and cloned into Tobacco Rattle Virus (TRV) RNA2 vector pJK037 using BsaI (New England Biolabs) and T7 DNA ligase (New England Biolabs). Constructs for localisation studies were assembled using Gibson Assembly® Cloning Kit (New England BioLabs® Inc). The entry vector pk2GW7.0, and mCherry from pSAT4A-mcherry-N1 (https://abrc.osu.edu/stocks/number/CD3-1081) were amplified by PCR with overhangs compatible for Gibson Assembly®. Potato lines overexpressing parakletos-mCherry or mCherry (EV, ctr) were created with Agrobacterium-mediated transformation[36]. The coding sequence of the potato Parakletos gene was analyzed for possible CRISPR targets and their numbers of off-targets using CRISPOR, and targets with the lowest numbers of potential off-targets were selected. The gRNA spacers were assembled into the Csy4 multi-gRNA vector pDIRECT_22C, using protocol 3A[37] to form the plasmid pDIRECT_22C_StParakletos. Potato *parakletos*-KO mutant lines were created and verified following the protocol of Kieu et al.[38]. Genomic DNA was prepared from WT and CRISPR-Cas9 mutants and used as a template in PCR using Phusion and two different Parakletos-primer combinations (Supplementary Table 1). Products were run on agarose gels. The PCR products were gel extracted and cloned into the pJET system (ThermoFisher). All selected plasmids were Sanger-sequenced (Eurofins).

### Protein fractionation and proteomic analysis

Potato plants were infiltrated with either *A. tumefaciens* or infiltration medium only (control) as described in[39,40]. The plants were sampled for protein extraction at 18 hpi (hours post infiltration). Eight biological replicates originating from two independent experiments were processed. Each sample consisted of two stabs from two potato leaflets at 18 hpi, corresponding to 100 mg fresh weight. Each sample was cooled on ice and put in a 1.5 mL Eppendorf tube with sea sand then processed with a Subcellular Protein Fractionation Kit for Tissues (ThermoFisher Scientific; Waltham, MA, USA, Cat. No. 87790) with minor modifications[39,40]. Briefly, proteins were consecutively extracted using ice-cold buffers and the final supernatants frozen at −80 °C until further use. The samples were homogenized in buffer 1, passed through a tissue and centrifuged at 500 g for 5 min at 4 °C. The pellet was washed once in buffer 1 and centrifuged. The pellet was re-suspended in buffer 2, vortex-mixed and incubated at 4 °C for 10 min with gentle mixing. After centrifugation at 3000 g for 5 min, the pellet was washed once with buffer 2. Buffer 3 was added to the resulting pellet, and the mixture was vortex-mixed and incubated for 30 min at 4 °C with gentle

mixing. After incubation, the sample was centrifuged at 5000 g for 5 min at 4 °C, and the supernatant cleared by re-centrifugation at 16,000 g for 10 min at 4 °C.

### Tryptic digestion and mass spectrometry

Proteins were separated on a 14% SDS-PAGE gel. The whole lane was removed and washed, and the proteins were digested using trypsin (Promega Trypsin Gold, Madison, WI, USA). The digests were desalted with C18-based spin columns (The Nest Group, Inc., Southborough, MA, USA)[41]. Samples were analyzed using a Q Exactive mass spectrometer interfaced with an Easy-nLC liquid chromatography system (both supplied by Thermo Fisher Scientific, Waltham, MA, USA). Peptides were analysed using NanoViper Pepmap pre-column (100 µm x 2 cm, particle size 5 µm, Thermo Fischer Scientific) and an in-house packed analytical column (75 µm x 30 cm, particle size 3 µm, Reprosil-Pur C18, Dr. Maisch) using a linear gradient from 7% to 35% B over 75 min followed by an increase to 100% B for 5 min, and 100% B for 10 min at a flow of 300 nL/min. Solvent A was 0.2% formic acid in water and solvent B was 80% acetonitrile, 0.2% formic acid. Precursor ion mass spectra were acquired at 70 K resolution and MS/MS analysis was performed in a data-dependent mode of the 10 most intense precursor ions at 35 K resolution and normalized collision energy setting of 27. Charge states 2 to 6 were selected for fragmentation and dynamic exclusion was set to 30 s.

### Peptide data analysis

Raw MS data were converted to Mascot generic file (mgf) format with ProteoWizard, as reported in Resjö et al.[39]. A *S. tuberosum* protein database from UniProt (www.uniprot.org), downloaded on 17 January 2017 and concatenated with a decoy database of equal size (random protein sequences with conserved protein length and amino acid distribution: 106,210 target and decoy protein entries in total) was generated using a modified version of the decoy.pl script from MatrixScience (http://www.matrixscience.com/help/decoy_help.html). This database was searched using the mgf files with Mascot version 2.3.01 in the Proteios software environment (https://proteios.org). Search tolerances were 7 ppm for precursors and 0.5 Da for MS/MS fragments. One missed cleavage was allowed and carbamidomethylation of cysteine residues was used as a fixed modification and oxidation of methionines as a variable modification. The *q*-values were calculated using the target-decoy method and the search results were filtered with a threshold peptide-spectrum match *q*-value of 0.01 to obtain a false discovery rate of 1% in the filtered list. Quantitative peptide analysis, was performed as reported earlier[40]. Normalized peptide data for 24946 identified peptide features was imported into InfernoRDN (https://omics.pnl.gov/software/InfernoRDN) and protein intensities were generated using the RRollup procedure. This resulted in quantitative data for 2178 proteins, which was then used in the further analysis. Differential expression was statistically analyzed using Limma[42], and all *p*-values were adjusted for multiple comparisons using the Benjamini-Hochberg procedure to calculate the corresponding *q*-values[43]. A *q*-value < 0.01 was required for a statistically significant differential expression.

### Agrobacterium-mediated transient expression and gene silencing in Nicotiana benthamiana

Transient expression was performedas reported earlier with minor modifications[44]. Briefly, *Agrobacterium tumefaciens* strain GV2260 harboring binary vector for Parakletos expression or empty vector was grown in LB medium supplemented with antibiotics at 28 °C overnight. Bacteria were pelleted by centrifugation and re-suspended in an infiltration buffer (10 mM MES, 10 mM MgCl$_2$, and 150 µM acetosyringone) to an OD600 of 0.1-0.2 and incubated at room temperature in the dark for 2 hours. Four to five weeks old *N. benthamiana* leaves were infiltrated using a 1 mL needless syringe. VIGS was performed in *N.*

*benthamiana*, as outlined in reference[45]. *Agrobacterium tumefaciens* strain GV2260 with binary TRV2:*Parakletos* or TRV2:*GFP* plasmid was mixed in infiltration buffer in a 1:1 ratio with bacteria carrying binary plasmid TRV1 at a final OD600 of 0.5. Two weeks old *N. benthamiana* plants were infiltrated and grown for another 3 weeks in a controlled growth chamber.

## Pathogen infection assays

*Phytophthora infestans* strain 88069 was grown on rye agar media plates and used for infection studies with *N. benthamiana* and *S. tuberosum*. Sporangia were collected after 12-14 days and their density was adjusted to 40000 per mL for infecting VIGS-treated *N. benthamiana* leaves, 25000 per mL for potato leaves and 60000 per mL for leaves subjected to *Agrobacterium*-mediated transient expression. Portions (10 μL) of sporangia suspension were placed on agro-infiltrated or VIGS-treated *N. benthamiana* leaves, and 25 μL on potato leaves. The infected plants were then maintained in a clear box with water to maintain 90-100% relative humidity. Infection severity was recorded 4–7 dpi by measurement of lesion diameter[46]. *P. infestans* biomass was quantified using a CFX96TM Realtime PCR system (Bio-Rad)[47]. Sporangia on 7-day-old leaves, of VIGS- and *Agrobacterium*-mediated over-expressing plants, were released by vortex-mixing the leaves in 5 mL water and then counted (sporangia/mL) using a haemocytometer. *Pseudomonas syringae pv. tomato DC3000 HopQ1* infection assay was performed as reported in Üstün et al[48]., bacteria was grown at 28 °C in King's B medium with rifampicin. Bacterial suspensions with an OD600 of 0.0001 were syringe-infiltrated into fully expanded leaves. The resulting bacterial colonies were counted after plate incubation for 1–2 days at 28 °C. *Dickeya dadantii* infection assay was performed as described in Kieu et al[49]., and results were recorded by measuring the size of lesions on each leaf at 5dpi. Infection assay for *Alternaria solani* strain 112 was conducted with results recorded by measuring the size of lesions at 5 dpi[24].

## Subcellular fractionation

Cell fractionation of *N. benthamiana* leaves, subjected to *Agrobacterium*-mediated transient expression of Parakletos-mCherry, was performed under very dim light or in darkness at 4 °C[50], with some modifications. Fresh leaves were ground in 2 mL icecold grinding buffer (50 mM Hepes/KOH (pH 7.5), 330 mM sorbitol, 2 mM EDTA, 1 mM MgCl₂, 5 mM ascorbate, 0.05% BSA, 10 mM sodium fluoride, 1 mM PMSF and 50 μM NaOV4), filtered through a prewetted 40μm filter and centrifuged at 2400 g for 4 minutes. The pellet was resuspended in 1 mL shock buffer (50 mM Hepes/KOH (pH 7.5), 5 mM sorbitol, 5 mM MgCl₂, 10 mM sodium fluoride, 1 mM PMSF and 50 μM NaOV4) and centrifuged at 7500 g for 4 min. The supernatant was re-centrifuged for 10 min at maximum speed and samples saved as "stroma fraction", while the thylakoid pellet was resuspended in 200 μL storage buffer (50 mM Hepes/KOH (pH 7.5), 100 mM sorbitol, 10 mM MgCl₂, 10 mM sodium fluoride, 1 mM PMSF and 50 μM NaOV4) and 30 μL samples were saved as "thylakoid fraction". Samples were run on 12% SDS gels and analyzed by western blot using an antibody against mCherry (see Immunoblotting).

## ROS measurements

ROS burst were detected in *N. benthamiana* and *S. tuberosum* leaf discs (0.125 cm²). Leaf discs were taken from leaves that had been pre-infiltrated (at 24 hpi) or from VIGS/KO plants. To minimize damage, leaf disks were cleaned with water and incubated overnight in wells of 96-well plates containing 200 μL water in the dark. The water was then replaced with 200 μL of solution containing luminol (17 mg/mL) and horseradish peroxidase (10 mg/mL) in sterile water, together with 1 μM synthetic flg22 peptide (QRLSTGSRINSAKDDAAGLQIA). The generation of reactive oxygen species (ROS) was monitored by recording light emitted through the oxidation of luminal in the following 60-

240 minutes using a GloMax® Navigator Microplate Luminometer. ROS bursts were defined as amounts of light released during this period, expressed in relative light units (RLU). Nitro blue tetrazolium (NBT) staining was carried out on potato leaves inoculated with or without *P. infestans*. Leaf dish samples (Φ 12 mm) collected 48 hpi were submerged into 0.2% NBT solution for 30 min and then replaced with 96% ethanol. Heat treatment (90 °C) was applied to remove chlorophyll pigment rapidly. Pictures were taken and quantified with ImageJ.

## Calcium burst measurements

Calcium burst was detected in *N. benthamiana* leaf discs[12]. For over-expression studies, *N. benthamiana* plants were transiently co-expressed with aequorin (AEQ) and the construct of interest. Plants used in the VIGS study were transiently expressed only with AEQ. Leaf discs (0.125 cm²) were taken and washed with water and incubated overnight in wells of 96-well plates containing 100 μL of 5 mM coelenterazine (Sigma) in the dark. The leaf discs were treated with a 1 mM flg22 and transient increase in calcium was recorded. Aequorin luminescence was measured with GloMax® Navigator Microplate Luminometer. Ca²⁺ bursts were defined as amounts of light released during this period, expressed in relative light units (RLU).

## Salt and drought stress

Stem cuttings from potato wild-type (WT) *parakletos* mutants (L19, 24, 72) were rooted on full strength MS-agar media for 7 days at 20 °C, and then transferred to hydroponics solution (4.3 g basal Murashige & Skoog salt (Duchefa biochemie) in 5 liter, pH 5.8 ± 0.2) with or without supplemented NaCl (60 mM) for salt stress. Likewise, for the drought stress experiment the 7-day rooted plantlets were equilibrated in the hydroponic solution before drought stress treatment in 20% PEG solution (polyethylene glycol 6000, Sigma). After 2 days, the PEG solution were removed, plant roots were washed 4 times with water and plants were keep in hydroponics solution for recovery. The plants were monitored daily to observe stress-induced changes. The fresh weight of WT and mutant plants was measured after 7 days recovery.

## Quantitative PCR

Total RNA was extracted from fresh leaf tissues using a Qiagen RNeasy mini kit (Qiagen, Hilden, Germany) following the manufacturer's recommendations. Nanodrop spectrophotometry was used to quantify RNA. A SuperScript III cDNA Synthesis Kit was used to synthesize cDNA (Thermofisher). The Platinum SYBR Green qPCR SuperMix kit (Thermofisher) was used to conduct qPCR on four biological and three technical replicates. The delta-delta Ct method was used to evaluate all qPCR data using CFX manager, with expression levels normalized to that of the housekeeping gene EF1. The primers to quantify the mRNA from NbEF1, NbICS1, NbPR1, NbrbohB, NbPTI5 and NbCAS are previously published[12,51], rest of the primer details are available in primers list (Supplementary Table 1).

## Microscopy

Confocal imaging: Agrobacterium GV2260 harboring relevant constructs were grown to a concentration of 0.8 OD600 and infiltrated in leaves (abaxial side) of 4-5 weeks old plants using a blunt 1 mL syringe. Three days post-infiltration, 0.5*0.5 cm leaf slices were cut, mounted in water and immediately examined using a Leica SP5 II confocal microscope equipped with a UV diode (405 nm), argon (488 nm) laser, HeNe (543 nm) laser and an N PLAN 50.0×0.75 BD or HCX PL APO lambda blue 20.0×0.70 IMM UV objective. The discs were sequentially scanned with the following settings, laser intensity 50% and: autofluorescence, 405 nm/emission 430–472 nm; GFP, excitation 488 nm/emission 503–542 nm; mCherry, excitation 543 nm/emission 568–633 nm. All scans were performed at room temperature (20–23 °C). The signals

were compared to those from empty vector-infiltrated leaves. The pictures were processed with LAS X (Leica Microsystems) software to generate overlays and enhance contrast/brightness. For flagellin induction 1-3 mL of 1 mM flg22 was infiltrated in the infiltrated areas of the tobacco leaves according to the time schedule. All pictures were treated identically.

Light microscopy: Leaf discs stained with NBT were examined using an inverted bright-field microscope. Images were captured at 400x magnification using a ZEISS Axiocam 503 microscope camera. The stained chloroplasts were analyzed using FIJI Image J software[52], where stained chloroplasts were selected and quantified by area from the captured images.

## Immunoblotting

Leaf material from transiently expressed constructs in *N. benthamiana* was completely grinded on dry ice using a mortar and pestle. Protein were extracted using cold (PEB) buffer (100 mM Tris, pH 6.8, 10%, glycerol, 0.5%, SDS, 0.1%, Triton X-100, 5 mM EDTA, 10 mM DTT, 5 μL protease inhibitor cocktail (cOmplete™, Sigma)). Samples were centrifuged 10 min at 10.000 rpm and the supernatants kept on ice. Samples were heated to 100 °C for 5 min in Laemmli Sample Buffer (BIO-RAD). Membranes (0.2 μm Nitrocellulose membrane (BIO-RAD)) were in 5% skim milk solution, and the primary antibody (BIO-RAD AHP2326, Polyclonal IgG Goat anti mCherry or Invitrogen- A-11120 GFP Monoclonal Antibody) was used in an overnight incubation 4 °C. After washing the secondary antibody, polyclonal rabbit anti-goat immunoglobulins/HRP (DAKO) was added and incubated 1 h under gentle shaking at RT. The membrane was washed three times and developed using SuperSignal® West Dura Extended Duration Substrate (Thermofisher) and visualised on a Universal Hood III (BIO-RAD) apparatus and analysed with the BIO-RAD Image Lab 5.2.1 software.

## Co-immunoprecipitation in *N. benthamiana*

Co-immunoprecipitation assay was performed as previously reported[53,54] with modifications. Briefly, three leaves of 5-week-old *N. benthamiana* plants were syring-infiltrated with *Agrobacterium tumefaciens* strain GV3101 expressing CPK16-GFP/CAS-GFP and mCherry/Parakletos-mCherry. Two days later, leaves were cut and ground in liquid nitrogen and homogenized in extraction buffer (50 mM Tris pH 7.5, 150 mM NaCl, 2.5 mM EDTA, 10% glycerol, 1% TritonX-100, 5 mM dithiothreitol, 1% plant protease inhibitor (Sigma, catalogue no. P9599)) (v/v). Proteins were solubilized at 4 °C with gentle agitation for 30 min before filtering through miracloth. The filtrate was centrifuged at 15,000 g for 20 min at 4 °C. An input sample was taken. For immunoprecipitation, 50 μL of GFP-Trap agarose beads (50% slurry, ChromoTek) were added and the mixture was incubated with gentle agitation for 4 h at 4 °C. Beads were harvested by centrifugation at 1,500 g for 2 min and washed three times in extraction buffer. Fifty microlitres of 4× elution buffer (NuPage) were added and incubated at 90 °C for 10 min. The samples were then spun at 13,000 g for 5 min before loading and running on SDS−PAGE and western blot detection with GFP antibody and mCherry antibody (see Immunoblotting).

## Phylogenetic analysis of plant Parakletos proteins

Protein sequences were identified by BLASTP using the NCBI (2022/10/30) and Uniprot (2022_11) databases. MEGA 11 was used for the alignment of full-length sequences using MUSCLE[55]. The evolutionary history was inferred by using the Maximum Likelihood method and JTT matrix-based model. The robustness of different nodes was assessed by bootstrap analysis using 1000 replicates.

## Field trials

Field trials of genetically modified potato plants were conducted at an established field station in Borgeby, southern Sweden, located at

55°45'12.6"N 13°03'10.7"E. Permission for these trials was granted by the Swedish Board of Agriculture (Dnr 4.6.18-01726/2020). Utilizing a randomized block design, the trials consisted of four replicates, with each replicate comprising a row of ten plants. We carried out fungicide treated and untreated trials in parallel. The trials followed the procedures outlined in reference[56]. Throughout the trials, there was adherence to the 'Environmental Code' (1998:808), the Swedish Board of Agriculture's Code of Regulations (SJVFS 2003:5), and the Regulation 2002:1086 regarding the deliberate release of GMOs into the environment.

Late blight disease incidence based on area under the disease progress curve (AUDPC) was calculated[57] from disease scoring from mid-June until 19th of august for year 2021 and from mid-June until 8th of august for year 2022. Both trails were sprayed against aphids with Fibro (paraffin oil) once per week, and once with the insecticide Teppiki. The fungicide-treated trial were treated with recommended doses of Revus (three times), followed by Ranman Top (two times), Infinito (two-four times) and Ranman Top (two times). All treatments were according to the manufacturer's recommendations. Fungicide treatment continued until the week before haulm killing. The data for fungicide treated plots in 2021, we regarded as difficult to conclude; thus, we repeated the fungicide-treated field trial in 2023. Stomatal conductance (gsw, mol m$^{-2}$ s$^{-1}$) and photosynthesis measurements, including the quantum yield of PSII (ΦPSII), were collected using the Licor (LI-600) instrument, which was used for field trials in 2022 and 2023. For the field trials in 2021, maximum quantum efficiency (Fv/Fm) was recorded with the FluorPen FP 100. Default program settings were used to estimate the stomatal conductance and photosynthesis measurements.

## Statistical analysis and data representation

All statistical analyses were performed by one-way ANOVA with the Minitab 18.1 software (https://www.minitab.com) or by two-sided Student's *t*-test (unequal variance) with Office Excel software. Box plots were plotted using BoxPlotR[58]. Details about the statistical approaches used can be found in the figure legends. Each experiment was repeated at least three times and data were represented as the mean ± SD or SE, as indicated; "n" represents number of samples. The centerline in a box plot indicates the median, + sign specifies mean value, the box borders delimit the lower and upper quartiles, and the whiskers show the highest and lowest data points. Adobe Illustrator 2022 software was utilized for figure presentation.

## Reporting summary

Further information on research design is available in the Nature Portfolio Reporting Summary linked to this article.

## Data availability

The MS proteomics data have been deposited at the ProteomeXchange Consortium via PRIDE partner repository with the dataset identifier PXD038421. All data supporting the findings of this work are available in the paper, Supplementary Information files, and repository platform. Source data are provided with this paper.

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

## Acknowledgements

We thank Jiorgos Kourelis for providing PJK037 gateway-compatible plasmid for virus-induced gene silencing. We thank Laura Medina and Eleanor Gilroy for providing CAS-GFP and TRV2:GFP, respectively. We also thank Daniel Hofius and Alan Collmer for generously providing *Pseudomonas syringae* pv tomato DC3000. We thank Dr. Fredrik Levander from the Department of Immunotechnology, Lund University and the Proteomics Core Facility at Sahlgrenska Academy for performing the peptide data analyses. This work was funded by following: Novo Nordisk Foundation NNF19OC0057208 (to E.A); Formas-The Swedish Research Council for sustainable development 2019-00512 and 2020-01211 (to EA and ML); Stiftelsen Lantbruksforskning R-19-25-282 (to EA); Carl Trygger Foundation CTS19:14 (to EA); The Nilsson-Ehle Endowments 40927 and 43352 from the Royal Physiographic Society of Lund (to M.A.Z.) and DFF/Independent Research Fund Denmark 1032-00399B (to BLP).

## Author contributions

M.A.Z., M.L., N.P.K., S.R., and E.A. jointly conceptualized the study, with M.A.Z. taking the lead. The experiments were carried out by M.A.Z., N.P.K., F.M.C., M.L., N.C.K., H.Y., S.J., J.M., R.V., B.L.P., S.R., and E.A. Data visualization and statistical analysis were performed by M.A.Z. and S.R. The manuscript was written by M.A.Z. with input from all authors and corrected by E.A. All authors read and approved the final manuscript.

## Funding

## Competing interests

M.A.Z., N.P.K., M.L., N.C.K., R.V., S.R., and E.A. are inventors on the patent application "Method of providing broad-spectrum resistance to plants, and plants thus obtained" (WO2022177484A1), where reduction of Parakletos expression is used for making plants more resistant. The other authors declare no competing interests.
