## [Peer Review File · Nature Communications]

Enhanced stress resilience in potato by deletion of ParakletosReviewers' Comments:

Reviewer #1:

Remarks to the Author:

This article describes a novel susceptibility protein, christened "Parakletos" by the authors, in potato thylakoids. The authors thoroughly tested the biochemical actions of this protein in *N. benthamiana* – an appropriate model – and then also in potato crops, over two years. Their findings that Parakletos knock-out potatoes are more resistant to many diseases and to some abiotic stresses could potentially have a large impact on agriculture. I found the paper to be well-written and thorough. The authors have a very convincing story here that merits publication almost without any revisions. I did find a few typographical errors that should be corrected – these are very minor:

Line 66: "it's" should be "its" – no apostrophe

Line 85: "that criteria" should be "the criteria"

Line 94: "have no" should be "has no"

Line 101: "what might" should be "which might" (in fact this whole sentence should be re-read by the authors. It seems to imply that Parakletos removes its own repression. Perhaps it's the way my brain is reading it)

Line 137: Should read "in crop plants"

Line 143: "These data" – not "this".

My only other suggestion is that the authors might add a short paragraph in the Discussion speculating on how or why a plant would retain a protein such as Parakletos if it only produces a penalty (increased disease and stress susceptibility), but without any obvious advantage – no yield penalty nor any other obvious side effects. This would have been interesting to read about.

Reviewer #2:

Remarks to the Author:

Authors found a new *S* gene, named Parakletos by proteomic analysis. Deficient of the gene confers resistance to broad-range of pathogens without any loss of yield of potatoes in the field, rather yield increased. These results are sound good, and could contribute to agriculture.

However, I have several questions describe below.

I confused the description about ROS production profiles induced by flg22 in Parakletos silencing plants.

Lanes 118-119; Authors described "A second ROS burst initiated after about 60 minutes, the cROS burst, was also stronger in Parakletos-silenced plants than in controls (Fig. 2c)". 60 min should be 120 min.

Lanes 204-209; Authors described "RBOHB was more strongly expressed in Parakletos-silenced plants than in controls (Fig. 2d), suggesting a mechanism by which Parakletos could affect plasma-membrane-related ROS production. However, elevated RBOHB transcript levels are unlikely to affect flg22-induced ROS bursts directly because transcript regulation occurs on a longer timescale than ROS bursts. In addition to its effect on the flg22-induced first ROS burst, Parakletos silencing also enhanced the second burst (Fig. 2c), which is associated with cROS".

Authors think that the second burst 120 min after flg22 infiltration in Fig. 2c is cROS. However, there is no data showing cROS, it is necessary to indicate RBOHB silencing does not affect the second burst, because a number of papers reported ETI induces not only chloroplastic ROS burst, but RBOH-mediated ROS burst accompanied by the gene expression. Moreover, luminol-based ROS assay can detect only apoplastic ROS generated by RBOH in the plasma membrane.

In this connection, NBT and DAB staining can detect both ROS bursts. Please observe the NBT-stained tissues in the Extended Data Fig.2c under the microscope to indicate cROS in chloroplasts. This is an important point, because authors speculate chloroplasts generate retrograde signals to control defense-related genes by flg22 in Parakletos deficient plants.

Reviewer #3:

Remarks to the Author:

This work identified a candidate gene, Parakletos, that enhances potato's broad-spectrum disease resistance and abiotic stress tolerance without affecting yield. Parakletos was discovered through quantitative potato proteomics and tested by functional assays. It negatively regulates ROS, calcium bursts and defence-related gene transcription. This gene could be a valuable tool for engineering resilient crops in the face of climate change and pathogen threats.

The work is rich in phenotype data, which support the crucial role of Parakletos in enhancing potato's fitness under diverse stress conditions. However, the molecular mechanism underlying Parakletos function remains elusive at this point. The authors need to address several major issues, such as the inconsistency between Fig. 1a and Fig. 1g-h, the unclear relationship between Parakletos and CAS, and the greatly varied yield between 2021 and 2022. Therefore, I recommend a major revision of the manuscript before publication.

1. In the abstract (line 34), the authors stated that: 'no yield penalty was evident in fungicide treated plots', which is confusing. The authors have noticed that deletion of parakletos did not cause a significant growth penalty (line 229), and a fungicide treatment would further protect the plants from diseases. Thus, why did the authors emphasize that KO lines had no effect under fungicide treatments?

2. The authors should provide a schematic showing the flow chart of proteomics sampling and analysis to help readers understand this part. They should also present data quality summary and general data analysis to validate the experiment, such as the number of proteins identified per sample, the experimental reproducibility, the statistical methods used, and the criteria for selecting differentially expressed proteins.

3. There is an inconsistency between Fig. 1a and Fig. 1g-h. Fig. 1a shows that M1CUF4-Parakletos is the third most increased protein in response to pathogen treatment, while Fig. 1g-h shows that Parakletos protein abundance declines following flg22 treatment. The authors should explain this discrepancy and provide additional data to support their claim. One possible reason is that the proteomics samples were collected at a later time point (18 hpi) than the flg22 treatment samples (120 min). The authors should detect Parakletos protein abundance for a longer period after plants are exposed to pathogens and PAMPs to see if there is a dynamic change in Parakletos expression.

4. The authors tested whether Parakletos acts in the same pathway as CAS, a thylakoid master regulator of plant immune responses. However, they did not provide enough background information about CAS, such as its function, regulation, and interaction partners. The authors should give a more detailed introduction about CAS and cite relevant literature to help readers understand the context and significance of their experiment.

5. Line 220, the authors stated that "Parakletos is functionally dependent on CAS and acts downstream of CAS (Fig. 2m-o)". However, this conclusion is not supported by the data. The fact that Parakletos and CAS double silenced plants act similarly to CAS single silenced plants suggests that Parakletos acts upstream of CAS, not downstream of CAS. The authors should revise their statement and provide a logical explanation for their hypothesis.

6. Fig. 3e and f show the effect of Parakletos on seedling growth under salt and drought stress. However, the initial status of the seedlings may greatly influence their growth rate. The authors should set a non-stress group and select seedlings that are in a consistent state to conduct different treatments. They should also measure and report the initial weight and length of the seedlings before the stress treatments.

7. Fig. 4c, the tuber yield varied greatly between year 2021 and 2022. For example, the yield of Désirée in 2021 was only ~13 ton ha⁻¹, whereas it dramatically increased to ~30 ton ha⁻¹ in 2022. The environmental factors that caused this large difference may also affect the real phenotype variations between WT and KO lines. A similar yield variation was also observed in Fig 4e.

8. Fig. 4c, disregarding the yield difference between two replicates, the resistance of KO lines was

higher than that of WT, but the KO lines still lost 20% to 50% of yield compared with fungicide treatments in Fig. 4e, suggesting this resistance was quite limited. The application of this gene for reducing fungicide use in the field might be more feasible than using this single gene for enhancing crop resilience.

9. Extended Data Fig. 5b shows a proposed model of Parakletos function in plants exposed to PAMPs. However, as mentioned in point 5, the relationship between Parakletos and CAS is not clear and may need further investigation. Moreover, the authors did not provide any evidence to show that Parakletos and CAS have a reciprocal interaction, as indicated by the arrow and bar between them. The authors should either remove or modify this part of the model, or provide experimental data to support it.

10. It would be interesting and give more depth to the research to clarify whether there is a direct and reciprocal interaction between Parakletos and CAS at the molecular level. The authors could use techniques such as co-immunoprecipitation, yeast two-hybrid, or bimolecular fluorescence complementation to test if Parakletos and CAS physically interact with each other. They could also use gene expression analysis, reporter assays to test if Parakletos and CAS regulate each other's transcription.

I hope these suggestions are helpful for improving the manuscript.

POINT BY POINT RESPONSE TO REVIEWER COMMENTS

We thank the reviewers for their critical comments, which have now been used to improve our manuscript (responses in red).

Reviewer #1 (Remarks to the Author):

This article describes a novel susceptibility protein, christened “Parakletos” by the authors, in potato thylakoids. The authors thoroughly tested the biochemical actions of this protein in *N. benthamiana* – an appropriate model – and then also in potato crops, over two years. Their findings that Parakletos knock-out potatoes are more resistant to many diseases and to some abiotic stresses could potentially have a large impact on agriculture. I found the paper to be well-written and thorough. The authors have a very convincing story here that merits publication almost without any revisions. I did find a few typographical errors that should be corrected – these are very minor:

Response: Thanks very much!

Line 66: “it’s” should be “its” – no apostrophe

Response: We have made this revision.

Line 85: “that criteria” should be “the criteria”

Response: We have made this revision.

Line 94: “have no” should be “has no”

Response: We have made this revision.

Line 101: “what might” should be “which might” (in fact this whole sentence should be re-read by the authors. It seems to imply that Parakletos removes its own repression. Perhaps it’s the way my brain is reading it)

Response: Thanks for the suggestion. Relating to this question, we have conducted new experiments based on the suggestions from Reviewer 3. This has also improved the readability of the statement, as recommended by Reviewer 1. We have made revisions in the main text, which are as follows: “ Furthermore, confocal microscopy and immunoblotting experiments indicated a rapid and transient reduction in *Parakletos* protein levels following flg22 treatment in *N. benthamiana* and potato (Figs. 1g, h; Supplementary Fig. 1e, f). A decline in *Parakletos* protein was also observed post *P. infestans* inoculation, followed by an increase (Supplementary Fig. 1g). Additionally, by immunoblotting, an elevation in *Parakletos* protein levels was detected 18 hours after *Agrobacterium* infiltration (Supplementary Fig. 1h), corroborating the proteomic data.

We have also changed the text in the discussion to: Therefore an attractive hypothesis is that Parakletos inhibits CAS function during biotic and abiotic stress, and that the transient reduction in Parakletos levels we observed relieves this repression.

Line 137: Should read “in crop plants”

Response: We have made this revision.

Line 143: “These data” – not “this”.

Response: We have made this revision.

My only other suggestion is that the authors might add a short paragraph in the Discussion speculating on how or why a plant would retain a protein such as Parakletos if it only produces a penalty (increased disease and stress susceptibility), but without any obvious advantage – no yield penalty nor any other obvious side effects. This would have been interesting to read about.

Response: We now have changed the sentences relating to this: A speculation could be that in nature, with higher degree of competition than in agricultural settings, Parakletos is important in fine tuning stress responses and thereby reducing cost of resistance. In contrast, in an agricultural setting, with generally more light and nutrients to each plant, increased transient defence activation could be beneficial, for example by the mutation of the negative regulator Parakletos.

Reviewer #2 (Remarks to the Author):

Authors found a new S gene, named Parakletos by proteomic analysis. Deficient of the gene confers resistance to broad-range of pathogens without any loss of yield of potatoes in the field, rather yield increased. These results are sound good, and could contribute to agriculture.

Response: Thanks very much!

However, I have several questions describe below.

I confused the description about ROS production profiles induced by flg22 in Parakletos silencing plants.

Lanes 118-119; Authors described “A second ROS burst initiated after about 60 minutes, the cROS burst, was also stronger in Parakletos-silenced plants than in controls (Fig. 2c)”. 60 min should be 120 min.

Response: We have made this revision.

Lanes 204-209; Authors described “RBOHB was more strongly expressed in Parakletos-silenced plants than in controls (Fig. 2d), suggesting a mechanism by which Parakletos could affect plasma-membrane-related ROS production. However, elevated RBOHB transcript levels are unlikely to affect flg22-induced ROS bursts directly because transcript regulation occurs on a longer timescale than ROS bursts. In addition to its effect on the flg22-induced first ROS burst, Parakletos silencing also enhanced the second burst (Fig. 2c), which is associated with cROS”.

Authors think that the second burst 120 min after flg22 infiltration in Fig. 2c is cROS. However, there is no data showing cROS, it is necessary to indicate RBOHB silencing does not affect the second burst, because a number of papers reported ETI induces not only chloroplastic ROS burst, but RBOH-mediated ROS burst accompanied by the gene expression. Moreover, luminol-based ROS assay can detect only apoplastic ROS generated by RBOH in the plasma membrane.

In this connection, NBT and DAB staining can detect both ROS bursts. Please observe the NBT-stained tissues in the Supplementary Fig.2c under the microscope to indicate cROS in chloroplasts. This is an important point, because authors speculate chloroplasts generate retrograde signals to control defense-related genes by flg22 in *Parakletos* deficient plants.

Reply: We thank the reviewer for these comments and suggestions. We acknowledge the oversight in our description of the detection capabilities of the luminol-based ROS assay and have revised the manuscript accordingly. In response to the reviewer's suggestion, we examined NBT-stained tissues from Supplementary Fig. 2c microscopically to identify cROS in chloroplasts. This analysis confirmed increased chloroplast staining in *parakletos*-knock-out plants (Supplementary Fig. 2d). We have updated the manuscript to include this information: "Moreover, microscopic examination of sections from NBT-stained leaves showed increased staining in the chloroplasts of *Parakletos*-KO lines, indicative of elevated chloroplastic ROS (cROS) (Supplementary Fig. 2d)."

Reviewer #3 (Remarks to the Author):

This work identified a candidate gene, *Parakletos*, that enhances potato's broad-spectrum disease resistance and abiotic stress tolerance without affecting yield. *Parakletos* was discovered through quantitative potato proteomics and tested by functional assays. It negatively regulates ROS, calcium bursts and defence-related gene transcription. This gene could be a valuable tool for engineering resilient crops in the face of climate change and pathogen threats.

The work is rich in phenotype data, which support the crucial role of *Parakletos* in enhancing potato's fitness under diverse stress conditions. However, the molecular mechanism underlying *Parakletos* function remains elusive at this point. The authors need to address several major issues, such as the inconsistency between Fig. 1a and Fig. 1g-h, the unclear relationship between *Parakletos* and CAS, and the greatly varied yield between 2021 and 2022. Therefore, I recommend a major revision of the manuscript before publication.

1. In the abstract (line 34), the authors stated that: 'no yield penalty was evident in fungicide treated plots', which is confusing. The authors have noticed that deletion of *parakletos* did not cause a significant growth penalty (line 229), and a fungicide treatment would further protect the plants from diseases. Thus, why did the authors emphasize that KO lines had no effect under fungicide treatments?

Response: We understand that the statement about fungicides in the abstract could be confusing, so we have removed it .

2. The authors should provide a schematic showing the flow chart of proteomics sampling and analysis to help readers understand this part. They should also present data quality summary and general data analysis to validate the experiment, such as the number of proteins identified per sample, the experimental reproducibility, the statistical methods used, and the criteria for selecting differentially expressed proteins.

Response: To explain the process of proteomics sampling and analysis more clearly, we have provided a schematic flow chart describing the procedure (Supplementary Fig.1a). We have also more information about statistical methods and criteria for selecting differentially expressed proteins. “Normalized peptide data for 24946 identified peptide features was imported into InfernoRDN (<https://omics.pnl.gov/software/InfernoRDN>) and protein intensities were generated using the RRollup procedure. This resulted in quantitative data for 2178 proteins, which was then used in the further analysis. Differential expression was statistically analyzed using Limma⁴², and all p-values were adjusted for multiple comparisons using the Benjamini-Hochberg procedure to calculate corresponding q-values⁴³. A q-value < 0.01 was required for a statistically significant differential expression.” Regarding the request for the number of proteins identified, we have elected not to include this. The reason for this decision is that this number is different for each replicate. Instead, we have added information about the total number of quantified peptides and proteins to the manuscript. For further details of each sample for example, we refer the readers to the files we have deposited in the PRIDE partner repository with the accession number PXD038421. Reviewers can access this data using the username reviewer_pxd038421@ebi.ac.uk and the password sR8L88Ju.

We have also now validated the proteomics findings with immunoblotting of Parakletos (Supplementary Fig. 1h).

3. There is an inconsistency between Fig. 1a and Fig. 1g-h. Fig. 1a shows that M1CUF4-Parakletos is the third most increased protein in response to pathogen treatment, while Fig. 1g-h shows that Parakletos protein abundance declines following flg22 treatment. The authors should explain this discrepancy and provide additional data to support their claim. One possible reason is that the proteomics samples were collected at a later time point (18 hpi) than the flg22 treatment samples (120 min). The authors should detect Parakletos protein abundance for a longer period after plants are exposed to pathogens and PAMPs to see if there is a dynamic change in Parakletos expression.

Reply: We appreciate the reviewer’s concern regarding the apparent inconsistencies between Fig. 1a and Fig. 1g-h. To address this, we conducted additional experiments to monitor the Parakletos protein abundance over an extended period post-exposure to pathogen and PAMP. In both *N. benthamiana* and a *Parakletos*-mCherry overexpressing potato line, we observed a transient decrease in Parakletos protein levels at 20 minutes post-flg22 treatment and no difference at later time points (Figs. 1g, h; Supplementary Fig. 1e, f). Contrarily, following exposure to *Phytophthora infestans*, a decline in Parakletos protein was observed at 18 hours post-infection, with an increase at 36 hours (Supplementary Fig. 1g). Additionally, Parakletos protein levels were elevated 18 hours after *Agrobacterium* infiltration (Supplementary Fig. 1h), supporting the findings from our proteomic analysis. These results have been incorporated into the manuscript as follows: “Furthermore, confocal microscopy and immunoblotting

experiments indicated a rapid and transient reduction in *Parakletos* protein levels following flg22 treatment in *N. benthamiana* and potato (Figs. 1g, h; Supplementary Fig. 1e, f). A decline in *Parakletos* protein was also observed post *P. infestans* inoculation, followed by an increase (Supplementary Fig. 1g). Additionally, by immunoblotting, an elevation in *Parakletos* protein levels was detected 18 hours after *Agrobacterium* infiltration (Supplementary Fig. 1h), corroborating the proteomic data.”

4. The authors tested whether Parakletos acts in the same pathway as CAS, a thylakoid master regulator of plant immune responses. However, they did not provide enough background information about CAS, such as its function, regulation, and interaction partners. The authors should give a more detailed introduction about CAS and cite relevant literature to help readers understand the context and significance of their experiment.

Response: Relevant information regarding CAS has now been added to the introduction section. “Chloroplast-mediated signalling pathways play key roles in generating ROS and Ca²⁺ bursts; biosynthesis of phytohormones, such as salicylic acid (SA); and retrograde signaling^{9,11}. The thylakoid-membrane-associated calcium-sensing receptor (CAS) plays a crucial role in these processes, including PTI-induced transcriptional induction, SA biosynthesis, and both bacterial and fungal resistance¹²⁻¹⁴. For example, pathogen effectors can modulate CAS activity in chloroplasts to suppress plant immune responses¹⁵. CAS also plays an important role in plant responses to abiotic stimuli¹⁴. Over-expression of CAS increases drought stress by preventing drought-induced reduction of photosynthetic efficiency¹⁶.”

5. Line 220, the authors stated that "Parakletos is functionally dependent on CAS and acts downstream of CAS (Fig. 2m-o)". However, this conclusion is not supported by the data. The fact that Parakletos and CAS double silenced plants act similarly to CAS single silenced plants suggests that Parakletos acts upstream of CAS, not downstream of CAS. The authors should revise their statement and provide a logical explanation for their hypothesis.

Response: We now have added CoIP experiment showing an interaction between Parakletos and CAS (Supplementary Fig. 2g). We have also removed the notion about downstream. We have made changes in the text as follows: “Co-immunoprecipitation experiments showed that Parakletos interacts with CAS, but not with the negative control CPK16^{G2A}, a mutant which localizes in chloroplasts (Supplementary Data Fig. 2g). These findings suggest that Parakletos and CAS are in the same protein complex.”

6. Fig. 3e and f show the effect of Parakletos on seedling growth under salt and drought stress. However, the initial status of the seedlings may greatly influence their growth rate. The authors should set a non-stress group and select seedlings that are in a consistent state to conduct different treatments. They should also measure and report the initial weight and length of the seedlings before the stress treatments.

Response: We appreciate the reviewer’s suggestion to account for the initial status of plant material in our stress experiments. Accordingly, we now included a non-stressed control group in the manuscript, and we present this data in Supplementary Fig. 4a-c for both salt and drought stress conditions. At the start of the experiments (day 0 in hydroponic conditions), both wild type (wt) and mutant plants were confirmed to have equivalent fresh weights (Supplementary Fig. 4a). After 7 days under hydroponic conditions (serving as the control for the salt stress

experiment), there was no significant difference in fresh weight between the control groups (Supplementary Fig. 4b). Prior to the drought stress, the fresh weight of wt and mutant plants was also comparable. After 14 days in hydroponic condition WT and mutant are not different (Supplementary Fig. 4c, left panel) then we divided the plants into two groups: control and stressed. After 23 days, there were no significant differences in fresh weight between wt and mutant seedlings under non-drought conditions (Supplementary Fig. 4c, right panel), confirming the initial uniformity of the experimental groups

7. Fig. 4c, the tuber yield varied greatly between year 2021 and 2022. For example, the yield of Désirée in 2021 was only ~13 ton ha⁻¹, whereas it dramatically increased to ~30 ton ha⁻¹ in 2022. The environmental factors that caused this large difference may also affect the real phenotype variations between WT and KO lines. A similar yield variation was also observed in Fig 4e.

Response: In our study, we had three years of yield data for fungicide-treated plots (Figure 4e). Two of these were similar (2022 and 2023), and 2021 was lower as pointed out. However, most importantly both year show a similar degree of difference between knockout and background potato. We think this actually indicate a robustness in the use of Parakletos KO plants. The general low yield in 2021 (compared to 2022 and 2023) is likely attributable to over-watering early that year. The year 2023 only fungicide treated experiments were done because we regarded the yield increase in non treated plots was so evident in the first two years, and we wanted to investigate any yield penalty in treated plots as that was not fully clear without that last year.

8. Fig. 4c, disregarding the yield difference between two replicates, the resistance of KO lines was higher than that of WT, but the KO lines still lost 20% to 50% of yield compared with fungicide treatments in Fig. 4e, suggesting this resistance was quite limited. The application of this gene for reducing fungicide use in the field might be more feasible than using this single gene for enhancing crop resilience.

Response: We agree with the reviewer's comments. We have revised the discussion to reflect that while the gene in question contributes to late blight protection, it should be considered part of a broader integrated pest management strategy rather than a standalone solution. The text now follows: “We emphasize that while *parakletos* deletion could contribute to late blight protection, it should be considered part of a broader integrated pest management strategy, rather than a standalone solution. This approach not only can extend the effectiveness of existing resistance genes but also provides a valuable defense against other stresses where options for fungicides or resistant genes are limited including abiotic stress conditions.”

9. Supplementary Fig. 5b shows a proposed model of Parakletos function in plants exposed to PAMPs. However, as mentioned in point 5, the relationship between Parakletos and CAS is not clear and may need further investigation. Moreover, the authors did not provide any evidence to show that Parakletos and CAS have a reciprocal interaction, as indicated by the arrow and bar between them. The authors should either remove or modify this part of the model, or provide experimental data to support it.

Response: We understand this criticism, so we decided to remove this model picture and it will be subject for future research.

10. It would be interesting and give more depth to the research to clarify whether there is a direct and reciprocal interaction between Parakletos and CAS at the molecular level. The authors could use techniques such as co-immunoprecipitation, yeast two-hybrid, or bimolecular fluorescence complementation to test if Parakletos and CAS physically interact with each other. They could also use gene expression analysis, reporter assays to test if Parakletos and CAS regulate each other's transcription.

Response: We have now added co-immunoprecipitation indicating that Parakletos and CAS are in a same protein complex (Supplementary Fig. 2g).

I hope these suggestions are helpful for improving the manuscript.

Response: We thank you for your comments and suggestions.

Reviewers' Comments:

Reviewer #1:

Remarks to the Author:

I am happy with the authors' responses to all of the reviewers' concerns, including my own. In my opinion, this article is now ready for publication.

Reviewer #2:

Remarks to the Author:

The authors revised all the requested parts.

I think the revised version of the manuscript is now good shape.

Reviewer #3:

Remarks to the Author:

This revised version of the manuscript includes changes in response to the previous comments, and I am pleased to see that most of the concerns have been addressed satisfactorily. Overall, the manuscript has been improved, and the research findings are presented more compellingly. I think that the manuscript is nearly ready for publication.